# Building a Better Defense: Expanding and Improving Natural Killer Cells for Adoptive Cell Therapy

**DOI:** 10.3390/cells13050451

**Published:** 2024-03-05

**Authors:** Andreia Maia, Mubin Tarannum, Joana R. Lérias, Sara Piccinelli, Luis Miguel Borrego, Markus Maeurer, Rizwan Romee, Mireia Castillo-Martin

**Affiliations:** 1Molecular and Experimental Pathology Laboratory, Champalimaud Centre for the Unknown, Champalimaud Foundation, 1400-038 Lisbon, Portugal; andreiaf_maia@dfci.harvard.edu; 2NK Cell Gene Manipulation and Therapy Laboratory, Division of Cellular Therapy and Stem Cell Transplant, Dana–Farber Cancer Institute, Harvard Medical School, Boston, MA 02115, USA; mubin_tarannum@dfci.harvard.edu (M.T.); sara_piccinelli@dfci.harvard.edu (S.P.); rizwan_romee@dfci.harvard.edu (R.R.); 3NOVA Medical School, NOVA University of Lisbon, 1099-085 Lisbon, Portugal; 4ImmunoTherapy/ImmunoSurgery, Champalimaud Centre for the Unknown, Champalimaud Foundation, 1400-038 Lisbon, Portugal; joana.lerias@research.fchampalimaud.org (J.R.L.); markus.maeurer@fundacaochampalimaud.pt (M.M.); 5Comprehensive Health Research Centre (CHRC), NOVA Medical School, Faculdade de Ciências Médicas (FCM), NOVA University of Lisbon, 1099-085 Lisbon, Portugal; luis.borrego@nms.unl.pt; 6Immunoallergy Department, Hospital da Luz, 1600-209 Lisbon, Portugal; 7I Medical Clinic, University of Mainz, 55131 Mainz, Germany; 8Pathology Service, Champalimaud Clinical Center, Champalimaud Foundation, 1400-038 Lisbon, Portugal

**Keywords:** innate immunity, NK cells, expansion methods, adoptive cell therapy, cytokines, feeder cells, clinical trials

## Abstract

Natural killer (NK) cells have gained attention as a promising adoptive cell therapy platform for their potential to improve cancer treatments. NK cells offer distinct advantages over T-cells, including major histocompatibility complex class I (MHC-I)-independent tumor recognition and low risk of toxicity, even in an allogeneic setting. Despite this tremendous potential, challenges persist, such as limited *in vivo* persistence, reduced tumor infiltration, and low absolute NK cell numbers. This review outlines several strategies aiming to overcome these challenges. The developed strategies include optimizing NK cell expansion methods and improving NK cell antitumor responses by cytokine stimulation and genetic manipulations. Using K562 cells expressing membrane IL-15 or IL-21 with or without additional activating ligands like 4-1BBL allows “massive” NK cell expansion and makes multiple cell dosing and “off-the-shelf” efforts feasible. Further improvements in NK cell function can be reached by inducing memory-like NK cells, developing chimeric antigen receptor (CAR)-NK cells, or isolating NK-cell-based tumor-infiltrating lymphocytes (TILs). Memory-like NK cells demonstrate higher *in vivo* persistence and cytotoxicity, with early clinical trials demonstrating safety and promising efficacy. Recent trials using CAR-NK cells have also demonstrated a lack of any major toxicity, including cytokine release syndrome, and, yet, promising clinical activity. Recent data support that the presence of TIL-NK cells is associated with improved overall patient survival in different types of solid tumors such as head and neck, colorectal, breast, and gastric carcinomas, among the most significant. In conclusion, this review presents insights into the diverse strategies available for NK cell expansion, including the roles played by various cytokines, feeder cells, and culture material in influencing the activation phenotype, telomere length, and cytotoxic potential of expanded NK cells. Notably, genetically modified K562 cells have demonstrated significant efficacy in promoting NK cell expansion. Furthermore, culturing NK cells with IL-2 and IL-15 has been shown to improve expansion rates, while the presence of IL-12 and IL-21 has been linked to enhanced cytotoxic function. Overall, this review provides an overview of NK cell expansion methodologies, highlighting the current landscape of clinical trials and the key advancements to enhance NK-cell-based adoptive cell therapy.

## 1. What Are Natural Killer Cells?

Natural killer (NK) cells are large granular lymphocytes representing 5–10% of circulating lymphocytes in healthy adults, playing a crucial role in recognizing and eliminating transformed and infected cells [1]. Distinguished by the absence of CD3 and CD19 and the presence of CD56, NKp46, and NKp80 expression, NK cells consist of two principal subsets: the immature or regulatory CD56^bright^ CD16^+/−^ cells and the mature or cytotoxic CD56^dim^ CD16^+^ cells (Figure 1A) [1,2,3]. The CD56^bright^ subset, more prevalent in the lymph nodes, secretes cytokines and chemokines in an inflammatory milieu, recruiting and modulating immune cells such as neutrophils, macrophages, T-cells, B-cells, and dendritic cells (Figure 1B) [4,5,6]. However, CD56^bright^ cells can be rapidly primed to acquire potent cytotoxic function upon cytokine activation [7]. In contrast, the CD56^dim^ subset exhibits higher direct cell cytotoxicity against tumor targets, mediating antibody-dependent cell cytotoxicity (ADCC) through the CD16 expression, which binds to the Fc region of IgG1 antibodies (Figure 1C) [8,9,10]. Furthermore, both NK cell subsets induce tumor cell apoptosis by expressing death ligands like FasL and TRAIL (Figure 1C) [11].

## 2. How Do NK Cells Modulate Their Immune Response?

NK cells exhibit a broad array of activating and inhibitory receptors, influencing their function [12]. Activating receptors include C-type lectin receptors, like CD94/NKG2C (binding to HLA-E) and CD94/NKG2D (binding to MHC-I chain-related molecules A/B (MIC-A/B)); natural cytotoxic receptors (NCR), including NKp30 (binding to hemagglutinin (HA), B7-H6, galectin-3, and glycosaminoglycans (GAGs)), NKp44 (binding to HA, GAGs, and NKp44L), and NKp46 (binding to HA, GAGs, and ecto-calreticulin); Fc receptor (FcγR, mediating ADCC); DNAM-1 (binding to poliovirus receptor (PVR) and Nectin-2); and activating killer-cell immunoglobulin-like receptors (KIR), such as KIR-2DS and KIR-3DS (recognizing MHC-I molecules) [12,13,14,15,16,17,18,19,20,21]. Inhibitory receptors include inhibitory KIR, such as KIR-2DL and KIR-3DL (recognizing MHC-I molecules), and C-type lectin receptors like CD94/NKG2A/B (binding to HLA-E) [12,13].

NK cell activation is mediated by a balance between activating and inhibitory signals (Figure 2). The “missing-self” hypothesis proposes that loss or downregulation of MHC-I molecules on tumor cells leads to decreased inhibitory signals, thus favoring NK cell activation [22]. In contrast, “stress-induced” activation occurs when stress ligands such as MIC-A/B are upregulated on infected or malignant cells, engaging activating receptors like NKG2D on NK cells [12,23]. In addition, NK cell function can be restrained by immune checkpoints, including NKG2A, TIM-3, TIGIT, and CD96 [24]. Although not highly prevalent, programmed death-1 (PD-1) expression has been observed in NK cells, thereby inducing dysfunction of the NK cells within the tumors with increased PD-L1 expression [25,26].

## 3. What Are the Major Advantages of NK Cells and Their Respective Sources for Adoptive Cell Therapy?

Chimeric antigen receptor (CAR)-T cells have significantly advanced cellular immunotherapy for cancer treatment [27,28,29,30]. Other T-cell-based approaches, including tumor-infiltrating lymphocytes (TIL) and T-cell receptor (TCR) T-cells, have also demonstrated promising efficacy, leading to the approval of several products by the FDA [30,31,32,33,34,35]. Nevertheless, T-cell-based therapies are associated with a relatively high risk of developing cytokine release syndrome (CRS), immune effector cell-associated neurotoxicity syndrome (ICANS), and hemophagocytic lymphohistiocytosis (HLH), resulting in prolonged cytopenia and risk of graph-versus-host disease (GVHD) in an allogeneic setting [36,37,38,39]. In contrast, NK/CAR-NK-cell-based products are not associated with these side effects, making them an attractive alternative [40]. In addition, the versatility of NK cell sources, including peripheral blood (PB), cord blood (CB), hematopoietic stem cells (HSC), and induced pluripotent stem cells (iPSC), increases their accessibility, supporting NK-cell-based products as a viable alternative to T-cells [41,42]. The major advantages and disadvantages of NK cell sources are described below (Figure 3).

### 3.1. Peripheral Blood (PB)

NK cells constitute approximately 5 to 10% of total lymphocytes in PB, exhibiting a mature phenotype with high cytotoxicity against tumors (Figure 3A) [1]. Within the PB-NK cells, the CD56^bright^ subset accounts for approximately 10%, while the remainder is mostly CD56^dim^ NK cells [43]. NK cells are commonly isolated from peripheral blood mononuclear cells (PBMC), obtained through leukapheresis, followed by a bead-based selection process [44]. Despite being readily available, using PB as a source for NK cells has several limitations, including relatively low cell numbers and donor-dependent variability [44].

### 3.2. Cord Blood (CB)

NK cells constitute around 23% of CB cells, with similar proportions of the CD56^bright^ and CD56^dim^ subsets found in PB (Figure 3B) [45]. CB-NK cells exhibit lower levels of adhesion molecules (CD2, CD11a, CD18, and CD54) and maturation receptors (KIR and CD57) while maintaining similar expression of the key cytotoxic molecules like granzyme B and perforin compared to PB-NK cells [46,47,48]. Although CB presents a limited number of NK cells per CB unit, recent efficient expansion strategies (described in the following section) have allowed the generation of several infusion products [49,50,51]. Alternatively, NK cells can be differentiated from the CD34^+^ HSC that are highly enriched in the CB and display many similarities to PB-NK cells, though with a lower inhibitory receptor expression [52,53,54,55].

### 3.3. Embryonic and Induced Pluripotent Stem Cells

Recent advances have allowed efficient differentiation of human embryonic stem cells (hESCs) and induced pluripotent stem cells (iPSCs) into highly functional NK cells (iNK cells), sharing many functional and phenotypic similarities (Figure 3C) [56,57,58,59]. iPSC-NK cells express CD56, DNAM-1, CD69, NKG2A/D, and NCR, like PB-NK cells [60,61,62]. However, the CD16 expression is lower, impacting their ability to mediate ADCC, while their KIR expression is variable, with certain cell populations expressing high levels of KIR and others not expressing [60,61,62,63]. Additionally, iNK cells generated from hESC expressed CD16 and KIR and lysed malignant cells by direct cell-mediated cytotoxicity and ADCC [55,62,64]. However, while iPSC-NK cells present a potential advantage for “off-the-shelf” manufacturing, their scale-up is somewhat more challenging than hESC-NK cells [59,65,66].

**Figure 3 cells-13-00451-f003:**
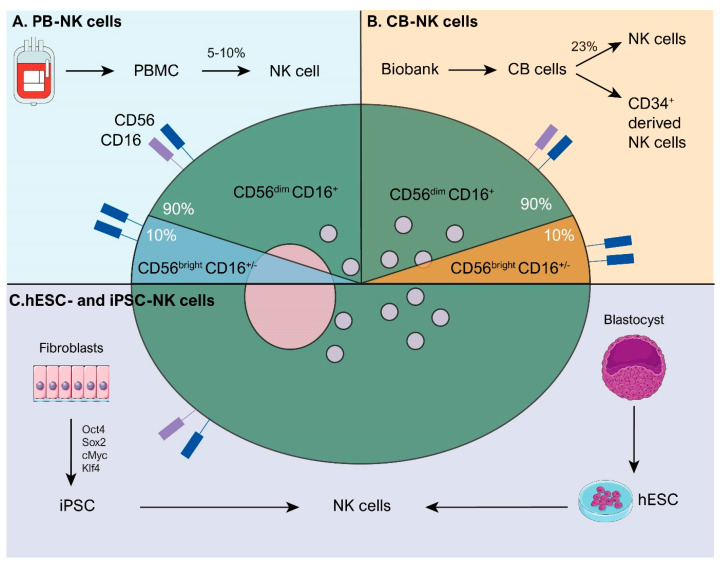
Sources of NK cells. (**A**) In PB-NK cells represent 5–10% of total lymphocytes, with approximately 90% being CD56^dim^ CD16^+^ NK cells and the remaining 10% comprising CD56^bright^ CD16^+/−^ NK cells. (**B**) CB-NK cells can be obtained from CB or by differentiating CD34^+^ cells through exposure to specific cytokines. NK cells comprise 23% of the total cells in CB, and the proportions of CB-NK cell subsets are similar to PBMC-NK cells. (**C**) Human embryonic stem cells (hESCs) and induced pluripotent stem cells (iPSCs) can generate NK cells with functional and phenotype similarities to PB-NK cells, although presenting a lower CD16 expression.

## 4. What Are the Major NK Cell Expansion Approaches?

Despite the availability of multiple sources for generating clinical-grade NK cell products, their low absolute numbers remain a limitation [67,68]. Several expansion methods have been developed to address this, with the major ones being summarized below [57,69,70,71,72,73,74,75,76,77,78,79,80,81,82,83,84,85,86,87,88].

### 4.1. Cytokine-Based Expansion

Cytokines such as IL-2, IL-12, IL-15, IL-18, and IL-21 play a crucial role in activating and regulating NK cell biology and, therefore, have been commonly used in various expansion methods [69,70]. IL-2 stimulates the *PI3K/AKT* pathway, promoting cell survival and proliferation via *mTOR* activation [89]. Sharing similar downstream pathways, IL-15 induces cell survival, activation, cytolytic activity, and cytokine production [90]. In contrast to IL-15, IL-2 also activates T-cells, including regulatory T-cells (Tregs) [91,92]. IL-12 enhances cytokine production, such as IFN-γ, and promotes cytotoxicity, while IL-18 sustains cell survival and exhibits co-stimulatory effects with other activation signals [93,94]. IL-21 supports NK cell survival and enhances cytotoxicity by upregulating granzymes and perforin molecules [95]. Various cytokine-based expansion protocols are highlighted in Table 1.

PBMC cultured with IL-2 achieved a 193-fold expansion of NK cells within 21 days [71]. IL-2, OK432, and an anti-CD16 monoclonal antibody (mAb) yielded a high NK cell purity of 76.9% with expansion folds ranging from hundreds to thousands, albeit with T-cell expansion [72]. Isolated NK cells cultured with IL-2 initially showed decreased NK cell counts by day 4, but recovered afterwards [73]. IL-15 demonstrated potential for NK cell expansion, while IL-21 triggered maturation and functionality, especially with short-term stimulation [74]. When expanded with IL-2, IL-15, anti-CD3 monoclonal antibody, tacrolimus, and dalteparin sodium, CB-NK cells achieved over 1700-fold expansion with more than 70% purity [75].

### 4.2. Feeder-Cell-Based Expansion

Feeder-cell-based expansion systems, typically involving immortalized or tumor cell lines in combination with cytokines, have become a common approach for expanding and activating NK/CAR-NK cells (Table 2) [76,77,78,96]. Using Epstein–Barr Virus-Immortalized Lymphoblastoid cell line (EBV-LCL) as feeder cells resulted in a 53-fold expansion of NK cells after one week with IL-21 stimulation and superior NK cell cytotoxicity [76]. In addition, CB-NK cells cultured with EBV-LCL cells showed an up to 6092-fold expansion after 35 days [77]. Furthermore, NK cells from patients with advanced cancer were cultured with NK-cell-depleted-PBMC from diseased and healthy donors [87]. A higher NK cell expansion rate was observed when using healthy donors’ feeder cells (300-fold) compared to diseased feeder cells (164.9-fold) [87].

Genetic alterations of feeder cells have allowed even higher NK cell expansion rates [57,79,80,81,82,83,84,85,97,98]. For instance, K562 cells expressing MICA, 4-1BBL, and soluble IL-15 induced a 550-fold cell expansion rate, increasing the NK cell percentage from 14.8% to 86.7% in PBMC [79]. NK cells cultured with K562 incorporating membrane-bound IL-15 (mbIL15) and 4-1BBL (K562-mbIL15-41BBL) induced a 277-fold expansion [80]. CB-NK cells cultured with K562-mbIL15-41BBL cells reached a 35-fold cell expansion and demonstrated superior cytotoxicity [81]. However, NK cells cultured with K562-mbIL15 cells displayed short telomeres [79]. In contrast, K562 cells overexpressing mbIL-21 (K562-mbIL21) promoted a massive log-phase NK cell expansion without evidence of senescence for up to 6 weeks [82]. Moreover, iNK cells cultured and expanded with K562-mbIL21-41BBL cells displayed 95.8% purity (CD3^−^CD56^+^ phenotype) [57]. A human B-lymphoblastoid cell line, 721.221, expressing mbIL21, induced superior NK cell expansion, purity, and cytotoxicity compared to NK cells cultured with K562-mbIL21 cells [83]. Additionally, K562-HLA-E feeder cells induced more than 10,000-fold expansion of NKG2C^+^ memory-like adaptive NK cells [97,98]. Furthermore, K562 cells expressing OX40 ligand (K562-OX40L) with IL-21 stimulation induced approximately 2000-fold NK cell expansion compared to 303-fold when using only K562 cells [84].

Feeder cells from autologous PBMC have also been successfully used to expand NK cells [78,86,87,88]. NK cells reached 62.7-fold expansion when cultured with irradiated autologous NK cell-depleted PMBC as feeder cells for 19 days [86]. In addition, CB-derived CD56^+^ cytotoxic cells cultured with IL-2 and irradiated autologous lymphocytes reached 156.3-fold expansion on day 26 [99]. Using autologous CD56-depleted feeder cells induced a 212-fold expansion of NK cells compared to 22.5-fold when using only IL-2 [88]. Additionally, culturing PBMC with irradiated autologous stimulated T-cells (FN-CH296) induced 90% purity and an NK cell median expansion of 4720-fold [78].

**Table 2 cells-13-00451-t002:** Analysis of NK cell expansion rates across diverse feeder-cell-based expansion methods, varying in source, media, feeder cells, duration, and culture material.

Ref.	Cell Source	Culture Factors	Feeders	Time (Days)	Culture Material	Results
[76]	PBMC-NK cells	TexMACS containing 5% HS type AB, 500 U/mL of IL-2, 100 ng/mL of IL-21, and feeder cells.	EBV-LCL cells (ratio 1:20).	7	N/A	NK cells reached 22-fold expansion in 1 week, which increased to 53-fold with IL-21.
[77]	CB-NK cells	X-VIVO 20 media with 10% heat-inactivated human AB serum, 500 IU/mL rhIL-2 and 2 mM GlutaMAX-1.	EBV-LCL cells.	Up to 40	T75 flasks.	CD3^−^ CD56^+^ cells reached a median of 6092-fold expansion.
[87]	Isolated NK cells	AIM-V medium supplemented with 5% HS, 1000 U/mL of IL-2 and OKT3.	NK cell—negative fraction of PBMC.	14	24-well plates.	The patient’s NK cells co-cultured with healthy donor feeder cells reached 300-fold expansion.
[79]	PBMC or purified NK cells	Media with 10 ng/mL IL-2.	Genetically modified K562 cells.	24	24-well plates.	K562-MICA-41BBL-IL-15 cells induced 550-fold NK cell expansion in 24 days.
[80]	PBMC or purified NK cells	RPMI-1640 media with 10 IU/mL human IL-2 and 10% FBS.	K562-mb15-41BBL cells.	21	24-well plates.	PBMC showed a 21.6-fold expansion of NK cells, while a 277-fold was reached in purified NK cells after 21 days.
[81]	CB-NK cells	RPMI-1640 media with 10% FBS and 10 IU/mL of recombinant IL-2.	K562-mbIL15-41BBL cells.	14	24-well plates.	K562-mbIL15-41BBL cells induced a 35-fold NK cell expansion.
[82]	PBMC	RPMI-1640 media with 50 IU/mL IL-2, 10% FBS, L-glutamine, and P/S.	K562-mbIL21 cells.	42	T75 flasks.	K562-mbIL21 cells induced a 47,967-fold expansion of NK cells by day 21.
[57]	iNK cells	B0 media supplemented with cytokines.	K562-IL21-4-1BBL cells.	42		At day 42, 98.5% of cells had a CD3^−^ CD56^+^ phenotype and reached 10^5^ to 10^6^-fold expansion.
[83]	PBMC	RPMI-1640 media with 10% FBS, 2mM L-glutamine, 100 U/mL P/S, 100 U/mL IL-2 and 5 ng/mL of IL-15.	221-mbIL21 cells.	20	G-Rex 6 Multiwell cell culture plates.	NK cells showed a 39,663-fold increase with 221-mbIL21 cells compared to a 3588-fold expansion with K562-mbIL21 cells.
[84]	PBMC	RPMI 1640 media with 10% FBS, P/S, 4 mmol/L of L-glutamine, and 10 U/mL of IL-2. After one week, IL-2 was increased to 100 U/mL, and 5 ng/mL of IL-15 was added. IL-21 was added at different concentrations.	K562-OX40L cells.	38	24-well plate	After four weeks, K562-OX40L cells and short exposure to IL-21 induced a 2000-fold expansion of NK cells.
[85]	Purified NK cells	AIM-V media with cytokines or feeder cells and 100 ng/mL OKT3 in the first culture cycle.	NK92-Neo2/15-OX40L cells.	21	N/A	NK92-Neo2/15-OX40L cells induced a 2180-fold increase of NK cells after 21 days without additional cytokines.
[86]	Isolated NK cells	CellGro SCGM with 5% HS, P/S and 10 ng/mL of OKT3. 200 U/mL of IL-2 was added alone or with 10 ng/mL of IL-15.	Autologous feeder cells.	19	Baxter LifeCell culture bags.	NK cells reached 62.7-fold expansion with IL-2 and 117-fold when IL-15 was added.
[99]	CB-NK cells	RPMI-1640 media with IL-2, 10% human AB, 1 mM L-glutamina, 10 U/mL Pen, and 0.01 mg/mL of streptomycin.	Autologous PBMC.	26	24-well plate.	CB-CD56^+^ cytotoxic NK cells reached 156.3-fold increased on day 26.
[88]	Isolated NK cells	TexMACS media with 5% human AB serum, 1000 U/mL of IL-2, and 10 ng/mL of OKT3.	Autologous CD56-depleted PBMC.	12	24-well plate, T25 and T75 flasks.	NK cells reached a 212-fold expansion with feeder cells, while only IL-2 showed a 22.5-fold expansion.
[78]	PBMC	GT-T507 with 1% plasma, IL-2, and OK-432.	FN-CH296 cells.	21–22	Flasks and culture bags.	A median of 4720-fold expansion was reached after 22 days with 90.96% purity.

Antibody (Ab), cord blood (CB), fetal bovine serum (FBS), human serum (HS), interleukin (IL), international unit (IU), natural killer (NK) cells, penicillin/streptomycin (P/S), peripheral blood mononuclear cell (PBMC), stem cell growth medium (SCGM), unit (U).

### 4.3. Culture Materials Used for NK Cell Expansion

Selecting appropriate culture materials, including flasks, bags, or bioreactors, is crucial in ensuring efficient NK cell expansion. Effector cells, particularly CD4^+^, CD8^+^, CD8^+^ CD56^+^ T-cells, and CD56^+^ NK cells cultured in bags and flasks, exhibited no significant differences in expansion, phenotype, or function over a 7-day period [100]. In contrast, NK cells reached a 530-fold expansion in bags compared to 1100-fold using flasks [101]. Culture in flasks carries the risk of exposure and contamination, which can be minimized in a GMP laboratory [101]. On the other hand, culturing in bags requires maintaining a certain cell concentration, and the gas exchange is restricted by the media’s volume, limiting the supply of nutrients and impacting NK cell proliferation [102]. Bioreactors are considered the most practical method despite being more expensive due to minimal hands-on time. However, bioreactors typically require higher starting cell numbers, and the expansion rate might be lower than other methods [101].

## 5. How Do Cell Culture Strategies Improve NK Cell Activity?

Although naturally cytotoxic, NK cells’ activity can be further improved during cell culture. The previous section described several expansion methods, while the current section outlines the changes in phenotype, persistence, and cytotoxicity that NK cells undergo during expansion.

### 5.1. Phenotype

NK cells display diverse activating and inhibitory receptors, and their expression often changes during cell culture [103]. Stimulation with IL-2, OK432, and anti-CD16 antibody upregulated CD16 and NKG2D, while IL-2 and IL-21 increased TRAIL, NKG2D, and DNAM-1 expression [72,76,78]. Combining IL-2 and IL-15 increased NKG2D and NKp44 levels, with no significant changes observed for NKp30, NKp46, and DNAM-1 [86]. NK cells cultured with PBMC (as feeder cells) showed a high frequency of RANKL, B7-H3, and HLA class II, particularly HLA-DR [88]. K562-MICA-41BBL-IL15 cells increased the expression of CD69, CD16, NKG2D, and CXCR3 on NK cells [79]. Despite higher NK cell expansion rates with K562-mbIL21 cells versus K562-mbIL15, NK cells exhibited a similar phenotype and cytotoxicity, maintaining donor KIR repertoire and showing high NCR, CD16, and NKG2D expression [82].

### 5.2. Telomere Length

Telomere shortening is a limiting factor for NK cell expansion [104]. Exposure to K562-mb15-41BBL cells limited proliferation due to telomere shortening [104]. To overcome this, NK cells overexpressing human telomerase reverse transcriptase (*TERT*) demonstrated an extended lifespan, maintaining a high percentage of cells in the S/G2 phase [105]. *TERT*-NK cells cultured with K562-mb15-41BBL cells showed prolonged proliferation for over a year, maintaining a normal karyotype and genotype [106]. NK cells in culture with K562 cells and IL-2 stimulation showed higher phosphorylation of STAT3, an activator of human *TERT* [73,107]. In addition, NK cells expanded with K562-mbIL21 cells, demonstrated increased telomere length, upregulation of *STAT3*, and reduced senescence [82].

### 5.3. Cytotoxicity

Stimulation with cytokines and feeder cells improves NK cell antitumor responses [108]. IL-2 and IL-12 promoted IFN-γ production, while IL-15 and brief IL-21 exposure boosted NK cell cytotoxicity, degranulation, and cytokine secretion [74,109]. Although K562-41BBL-IL15 cells promoted NK cell cytotoxicity and IFN-γ production, using K562-MICA-41BBL-IL15 induced a stronger effect [79]. Additionally, K562-mbIL21-41BBL cells also improved IFN-γ and TNF-α production [82]. NK-92-OX40L cells secreting neoleukin-2/15 (Neo-2/15) increased NK cell cytotoxicity against tumor cells [84].

Expansion strategies influence NK cell phenotype, function, and cytotoxicity, determining the efficacy of NK-cell-based therapies.

## 6. Which Other Strategies Improve NK Cell Antitumor Response?

Despite improvements in NK cell expansion and cytotoxicity during cell culture, their clinical application is hampered by major challenges, including short half-life, limited tumor infiltration, and low *in vivo* persistence [110]. Key strategies to further enhance NK-cell-based immunotherapeutic approaches include stimulation of memory-like NK cells, genetic manipulation including CAR-NK cells, and, more recently, TIL-NK cells [111,112,113].

### 6.1. Memory-like NK Cells

NK cells can acquire memory-like properties in response to cytokine stimulation, particularly IL-12, IL-15, and IL-18, generating what is called the cytokine-induced memory-like (CIML) NK cells or, in the context of viral infections like cytomegalovirus (CMV), inducing what is called the adaptive NK cells [112]. CIML NK cells undergo transcriptional, epigenetic, and metabolic reprogramming, leading to increased proliferation, cytotoxicity, and long-term persistence in mouse models and patients [114,115,116]. Upregulation of phosphorylated STAT5 and demethylated conserved noncoding sequence 1 (CNS1) was found in memory-like NK cells [114,117]. In addition, increased expressions of IL-2Rα (CD25), nutrient transporters, including transferrin receptor (CD71), amino acid transporter (CD98), and glucose transporters (GLUT1 and GLUT3) were observed [118,119,120]. These cells demonstrated enhanced function, proliferation, and *in vivo* persistence against cancer cells [112,121,122]. On the other hand, under CMV reactivation, adaptive NK cells demonstrate inferior NKG2A, NKp30, and NKp46 expression while increasing NKG2C, KIR, and CD57 markers [123,124]. Adaptive NK cells are characterized by a memory-like phenotype and increased cytotoxicity, producing more IFN-γ molecules with long-term persistence, making them attractive for immunotherapy [115,123].

### 6.2. Chimeric Antigen Receptor (CAR) Technology

CAR therapy involves modifying immune cells to express a synthetic receptor binding to a tumor-associated antigen like CD19, expressed by B-cells (both normal and B-cell lymphomas), and BCMA, expressed by normal and malignant plasma cells [29,125,126]. While T-cells are the most widely used immune cell for CAR therapy, they present several limitations, such as the alloreactivity and GVHD risks, making autologous T-cells a preferential choice [38,127]. In contrast, CAR-NK cells, derived from allogeneic sources, provide an “off-the-shelf” product without inducing GVHD, avoiding the massive release of cytokines and neurotoxicity [51,128].

CD19-CAR-CIML NK cells demonstrated enhanced IFN-γ production, degranulation, and specific killing against NK-resistant lymphoma cell lines [129]. Recently, CD19-CAR-CB NK cells expressing IL-15 demonstrated safety and efficacy in CD19^+^ B-cell malignancies [40]. In this study, 73% of patients had a response to the treatment without toxicities observed, including CRS, GVHD, and neurotoxicity [40,51]. CAR-CIML NK cells targeting a neoepitope generated in nucleophosmin-1 (*NPM1*)-mutated acute myeloid lymphoma (AML) displayed potent activity, improving AML outcomes in xenograft models [130]. Various strategies have recently been explored to improve CAR-NK cells, including *CISH* knockout, promoting metabolic fitness, cell expansion, antitumor cytotoxicity, IL-15-mediated *STAT* signaling, and superior NCR expression [131,132,133]. In addition, enhanced NK cell functions against glioblastoma through αv integrin blockade, TGF-β inhibition, and CRISPR gene editing of the *TGFBR2* gene were also reported [134]. CD38 knockout NK cells overcome the daratumumab-induced fratricide, improving AML and multiple myeloma (MM) treatment efficacy [135]. Overexpression of CXCR1 in NKG2D-CAR-NK cells enhanced their migration toward *in vitro* and *in vivo* tumors [136]. Similarly, overexpression of CXCR4 or CCR7 in NK cells improved migration and infiltration into specific tissues, reducing tumor burden and extending survival in mice [137,138].

### 6.3. Tumor-Infiltrating Lymphocytes (TILs), including TIL-NK Cells

TILs primarily comprise T-cells that have migrated into a tumor [139]. TIL therapy has shown significant clinical results, particularly in melanoma patients [140,141,142]. Furthermore, ongoing studies in breast and colorectal cancers highlight the potential of TILs as a source of antigen-specific immune cells [113,143,144]. Recent studies have emphasized that TILs include NK cells (TIL-NK cells), associated with improved prognosis in multiple malignancies [145,146,147,148,149,150]. In lung cancer, TIL-NK cells are predominantly CD56^bright^ perforin^low^, exhibiting lower cytotoxicity but with similar cytokine production compared to the PB-NK cells [151]. In soft tissue sarcoma, NK cells represent around 20% of the TIL population [152]. In comparison, less than 0.5% of TIL-NK cells are found in pancreatic ductal adenocarcinoma (PDAC), attributed to the low expression of the chemokine receptor CXCR2 [153]. Transgenic expression of CXCR2 facilitates NK cell infiltration, although proliferation is limited in the hypoxic TME [154]. CCR7 expression, involved in lymphocyte migration to lymph nodes, remains unchanged, while CXCR3, mediating NK cell recruitment to tumor sites, is enhanced in expanded NK cells [79]. Additionally, CXCL9 and CXCL10 expression in the TME and IL-15 stimulation promote the recruitment of NK cells and cytotoxic CD8^+^ T-cells via CXCR3 into the tumors [155].

While until now, no method has been explicitly published for the expansion of TIL-NK cells, it is crucial to consider the promising prospects of TIL-NK-cell-based therapies in cancer [149,156]. Despite the limited cell numbers of TIL-NK cells, their superior tumor-infiltration capacities and association with improved overall survival of cancer patients underscore significant potential [149,156]. Additionally, the recent FDA approval of Amtagvi, an autologous TIL therapy for advanced melanoma in adults, emphasizes the promising potential of TIL therapy as a treatment option for cancer [157,158].

## 7. Which Are Currently the Major NK-Cell-Based Clinical Trials?

The current clinical application of NK cells includes autologous and allogeneic NK-cell-based approaches [159]. So far, the NK cell clinical trials have predominantly focused on patients with hematological diseases, though promising data from recent preclinical studies strongly support their evaluation in solid tumors [160,161,162]. The following section summarizes key clinical trials, highlighting the advantages of expanded NK cells.

### 7.1. Autologous NK Cells

In the autologous setting, NK cells have been safely infused and expanded *in vivo* with IL-2 administration; however, their efficacy has been limited [78,163,164]. In a phase I trial, autologous NK cells expanded with K562-mbIL15-41BBL cells reported stable disease combined with trastuzumab in 6 of 19 patients with HER2-positive malignancies [163]. Infusion of activated autologous NK cells into MM patients post-autologous HCT also supported the broader feasibility of this therapy [165]. Despite the prolonged survival of *ex vivo* IL-2-activated autologous NK cells in preclinical studies, no clinical responses were observed in patients with metastatic melanoma or renal cell carcinoma [166]. Their minimal clinical activity can be due to a lack of KIR/ligand mismatch in the autologous tumor cells and/or due to their limited *in vivo* persistence after adoptive transfer [166]. These challenges represent a significant hurdle for autologous NK-cell-based therapy [166]. Several ongoing clinical trials are evaluating autologous NK cell therapy for hepatocellular carcinoma (NCT06044506) and MM in combination with low IL-2 (NCT04634435).

### 7.2. Allogeneic NK Cells

The use of allogeneic NK cells has been associated with inducing remission and preventing relapse in AML and MM patients [167]. Using KIR-mismatched donor NK cells after haploidentical HCT demonstrated a significantly reduced risk of relapse in high-risk AML [168,169]. Subsequently, allogeneic NK cells from unrelated healthy donors were assessed in advanced lymphoma and solid tumors [160]. Among 17 patients, 47.1% showed stable disease, highlighting the safety and the potential efficacy of administrating random-donor allogeneic NK cells and thus expanding cell donor options [160]. In another study, the use of haploidentical PBMC (CD3^+^ T-cell-depleted and NK-cell-enriched) was safe and induced complete response (CR) in 5 of 19 poor-prognosis AML patients [170]. Similarly, IL-2-activated allogeneic NK cells combined with anti-CD20 mAb yielded responses in 14 of 15 relapsed/refractory CD20^+^ lymphoma patients in a phase II clinical trial [171].

While haploidentical NK cell infusion induced remissions, the presence of Tregs may have contributed to their diminished efficacy [172]. Depletion of host Tregs with IL-2-diphtheria fusion protein improved the efficacy of haploidentical NK cell therapy, resulting in higher donor NK cell expansion in relapsed-refractory AML patients [172]. Substituting IL-2 with IL-15 showed promising results, with 36% of patients exhibiting robust *in vivo* NK cell expansion and 32% achieving CR, avoiding Treg stimulation [173]. In a phase II trial, IL-15 administered subcutaneously (SC) resulted in NK cell expansion in 27% of the patients, and 40% achieved remission [173]. However, while IL-15 improved *in vivo* NK cell expansion and remission rates, it was also associated with previously unreported CRS after SC administration [173]. Moreover, IL-15 superagonist complex ALT-803 was well tolerated, stimulating NK and CD8^+^ T-cells without increasing Tregs [174].

Considering the high risk of relapse after allogeneic HCT, donor-derived CIML NK cells are attractive in myeloid malignancies that have relapsed after haploidentical HCT [175]. A first-in-human phase I trial with donor CIML NK cells in relapsed or refractory AML reported four of nine patients achieving CR and one achieving morphologic leukemia-free state (MLFS), resulting in an overall response rate of 55% and a CR rate of 45% [122]. Additionally, CIML NK cell infusion led to rapid 10- to 50-fold expansion *in vivo*, sustained over months, supporting CIML NK cells as a platform for post-transplant relapse myeloid disease treatment [175]. These findings highlight the importance of expanding and stimulating NK cells before infusion, whether as CIML or conventional NK cells, to improve immunotherapy efficacy.

### 7.3. Allogeneic CB-NK Cells

In a first-in-human trial, CB-NK cells, expanded with IL-2 and K562-mbIL21 cells, were infused in MM patients receiving high-dose chemotherapy and autologous HCT [176]. This trial demonstrated safety and efficacy, with 10 patients achieving at least a good partial response, including 8 near CR [176]. Additionally, in recurrent ovarian carcinoma, CB-NK cells exhibited safety and *in vivo* expansion capacity [177]. The “off-shelf” CB-NK cell product, oNKord, has obtained approval for AML patients [178]. The phase I trial demonstrated safety and efficacy in elderly AML patients, ineligible for allogeneic HCT, while an ongoing phase II trial is evaluating oNKord in patients with minimal residue disease (MRD) (NCT04632316) [178]. Recently, a phase I/II trial of CB-NK cells expressing CD19-CAR and IL-15 was evaluated in patients with CD19^+^ B-cell malignancies [40,51]. The 1-year overall survival (OR) and progression-free survival were 68% and 32%, respectively, with patients achieving OR correlated with higher levels and longer persistence of CAR-NK cells [40]. No notable toxicities were observed, including CRS and GvHD [40].

## 8. What Is the Future of NK-Cell-Based Therapy?

While NK cells hold great promise, challenges remain, including low *in vivo* persistence and proliferation capacity, which can compromise their effectiveness in cancer therapy [179]. This review aims to elucidate several strategies used to improve NK cell proliferation and antitumor function, either through expansion methodologies, cytokine stimulation, or genetic modifications. Diverse culture methods are explored using various cytokines (such as IL-2, IL-12, IL-15, IL-18, and IL-21) and feeder cells (including genetically modified or nongenetically modified cell lines, as well as autologous or allogeneic cells), with emphasis on their effects on NK cell expansion, phenotype, telomere length, and cytotoxic activity. Notably, stimulation with IL-2 and IL-15 has been shown to promote NK cell expansion, while IL-12 and IL-21 have been associated with enhancing cell cytotoxicity [71,72,73,74]. In addition, using genetically engineered feeder cells, like K562-mbIL21 cells, has demonstrated remarkable *ex vivo* NK cell expansion capacities [82,180].

Furthermore, the generation of memory-like NK cells, characterized by superior IFN-γ production and cytotoxicity, represents another strategy to improve *in vivo* persistence, providing a sustained and potent functional alternative [122]. On the other hand, genetic manipulation of NK cells to develop CAR-NK cells has shown promising results, enhancing NK cell targeting without inducing severe side effects [40,51,128,129,136,137]. Moreover, exploring TIL therapy, particularly TIL-NK cells, presents a compelling alternative to improve tumor treatment, given their association with improved overall survival in cancer patients [149,156]. Recently, FDA approval of an autologous TIL therapy for advanced melanoma has underscored the promising antitumor potential of TILs [157,158].

Despite the promising outcomes in cancer, especially in hematological malignancies, translating NK-cell-based therapies in solid tumors faces significant challenges, with poor tumor trafficking and highly immunosuppressive TME as major barriers for both NK- and non-NK-cell-based cellular therapy approaches [110]. The TME can induce NK cell dysfunction and exhaustion through various mechanisms, including suppressive immune or nonimmune cells (e.g., Tregs and myeloid-derived suppressor cells (MDSC)), cytokines like TGF-β, overexpression of inhibitory ligands (e.g., HLA-E), and downregulation of activating ligands [181]. Several strategies are being evaluated to overcome these challenges, including exploring TIL’s superior tumor infiltration capacity, and developing novel CAR-engineered cells [182,183,184]. Engineered NK cells with synthetic receptors, sustained cytokine production, safety mechanisms like drug-inducible suicide genes, or “on/off switches” through small molecule administrations promise great potential [185]. Targeting cancer and other cells in the TME that help tumor growth, like cancer-associated fibroblasts (CAFs), can potentially improve NK-cell-based approaches.

Furthermore, the most effective NK-cell-based immunotherapy may involve combining them with other immune cells, such as CAR-T cells, TIL, or CAR-macrophages, taking advantage of the unique strengths of each approach to enhance tumor control significantly. Additionally, combining NK cell therapies with immune checkpoint inhibitors might be a great strategy, especially considering the dysfunction and exhaustion markers expressed by NK and T-cells within the TME, such as CD161, TIGIT, and CD96 [152]. Lastly, there is a huge interest in developing the *in vivo* “arming” of NK and other immune cells to mitigate labor-intensive and expensive adoptive cell therapies [186]. Recent progress in RNA-based approaches combined with nanoparticle-based technology makes *in vivo* modulation of the immune cells feasible. However, challenges like low transfection efficacy, short RNA half-life, and limited cell specificity remain the major barriers to these efforts [187].

In summary, this review offers insights into strategies aimed at enhancing NK cell function through expansion methodologies or genetic modifications, elucidating their impact on NK cell phenotype, proliferation, and cytotoxicity. In addition, this review highlights the current clinical trial landscape and key advancements aimed at further enhancing NK-cell-based adoptive cell therapy.

## Figures and Tables

**Figure 1 cells-13-00451-f001:**
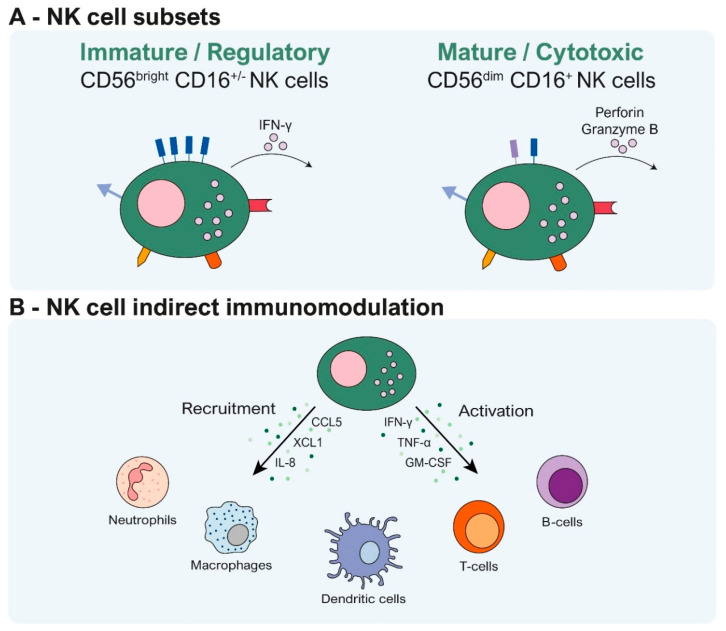
Representation of the NK cell subsets and NK-cell-mediated immunomodulation and cytotoxicity mechanisms. (**A**) NK cells are classified by CD56 and CD16 surface markers, including regulatory or immature NK cells (CD56^bright^ CD16^+/−^), primarily releasing IFN-γ molecules, and cytotoxic or mature NK cells (CD56^dim^ CD16^+^), mainly releasing perforin and granzyme B molecules. (**B**) Activated NK cells, in particular regulatory NK cells, release immunomodulatory mediators such as CCL5, XCL1, IL-8, IFN-γ, TNF-α, and GM-CSF within the TME, recruiting and activating other immune cells, including neutrophils, macrophages, dendritic cells (DC), T-cells, and B-cells. (**C**) Upon activation, NK cells exert cytotoxic effects through FasL and TRAIL-induced apoptosis, degranulation of perforin and granzyme molecules, and through antibody-dependent cell cytotoxicity (ADCC) via CD16, particularly cytotoxic NK cells, binding to antibodies on tumor cells, inducing apoptosis.

**Figure 2 cells-13-00451-f002:**
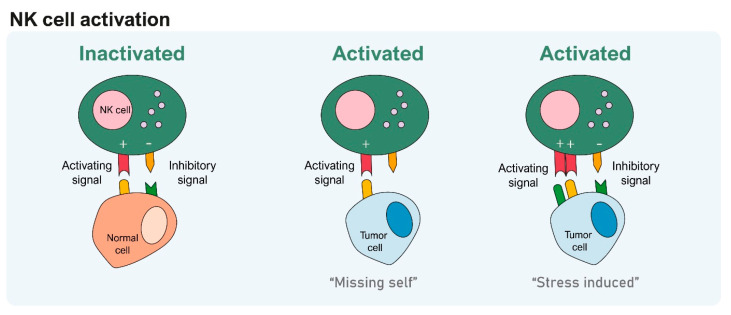
NK cell activation mechanisms. A balance of inhibitory and activating signals regulates NK cells. This balance is maintained when NK cells encounter normal cells, resulting in NK cell inactivity. However, certain conditions can activate NK cells, such as recognizing “missing self” when MHC-I molecules, serving as negative signals, are absent. Additionally, NK cells can be activated through “stress-induced” mechanisms when stress ligands, acting as positive signals, are overexpressed in stressed cells, including tumor- or viral-infected cells.

**Table 1 cells-13-00451-t001:** Analysis of NK cell expansion rates across diverse cytokine-based expansion methods, varying in source, media, duration, and culture material.

Ref.	Cell Source	Culture Factors	Time (Days)	Culture Material	Results
[71]	PBMC	CellGro SCGM with 5% HS, anti-CD3 Ab for the first 5 days with 500 U/mL of IL-2.	21	6-well plates and T25 flasks.	CD3^−^ CD56^+^ cells reached 193-fold expansion.
[72]	PBMC	Serum-free medium with 700 IU/mL IL-2, 0.01 KE/mL OK432, 10% human plasma, and an anti-CD16 Ab.	21	Flask and culture bag.	Fold expansion ranges from 637 to 5712, with a purity of 76.9%.
[73]	Isolated NK cells	X-VIVO 10 media with 5% heat-inactivated human FFP and 1000 U/mL of rhIL-2.	12	GMP-grade VueLife culture bags.	The mean expansion rate of NK cells was 4-fold, while two donors reached 30-fold.
[74]	Isolated NK cells	X-VIVO 10 media with 5% heat-inactivated human FFP, P/S, 100 or 1000 U/mL of IL-2, 10 ng/mL of IL-15, 25 ng/mL of IL-21 or combinations of those.	42	T25 flasks.	IL-15 induced NK cell expansion, while IL-21 triggered NK cell maturation and functionality.
[75]	CB-NK cells	SCGM with 5% human AB serum, IL-15, IL-2, anti-CD3 ab, tacrolimus, and dalteparin sodium.	20	24-well plates and T25 flasks.	1700-fold expansion with 72.8% purity.

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
