# Peer review of "Building a Better Defense: Expanding and Improving Natural Killer Cells for Adoptive Cell Therapy"

_cells, 2024, doi:10.3390/cells13050451_

Round 1
Reviewer 1 Report
Comments and Suggestions for Authors
Natural killer (NK) cells are one of the promising adoptive cell therapy platforms for cancer treatment. NK cells have some advantages over T-cells, including MHC-I-independent tumor recognition and a low risk of toxicity. However, there are some disadvantages as well. This review provides an overview of the several strategies to overcome limited in vivo persistence, reduced tumor infiltration, and low absolute NK cell numbers.
The topic is quite original and relevant in the field. This review provides detailed information about the function of NK-cells, the benefits of their application in treatment, all types of sources of these cells and the differences between them. For any practicing researcher this review offers methodological details for the cultivation and expansion of NK-cells. Then, the authors present all current strategies for the increase in the efficiency of cancer treatment with NK-cells. All this information is highly needed for the researchers.
The data are detailed and explicit compared with other published material.
The references are appropriate, the number of the articles reviewed is representative.
The tables and figures are useful and descriptive.
I think, this manuscript is of high-quality and should be published.
As a minor comment I can only ask for the more obvious conclusion part in the end of the article. It would improve the text if there were not only wors about the future.
Author Response
We thank this reviewer for the positive comments regarding our manuscript. Following the minor comment, we have now included a new paragraph of Conclusions to finalize the manuscript that we hope the Reviewer will find appropriate and in line with the manuscript.
Reviewer 2 Report
Comments and Suggestions for Authors
Review report
“Building a Better Defense: Expanding and Improving Natural Killer Cells for Adoptive Cell Therapy” review article by Andreia Maia is a summary of expansion protocols and harnessing of NK cells from multiple sources for adoptive cell therapy. Authors discuss the pros and cons associated with adoptive cell therapy using NK cells. This review address some of the limitations of NK cell based adoptive immunotherapy and provide insight into potential strategies for circumventing these limitations. Authors describe strategies for “massive” NK cell expansion using NK sensitive target cells K562 expressing membrane IL-15 or IL-21 with additional activating ligands like 4-1BBL and highlight the feasibility of multiple cell dosing and “off-the-shelf” availability for cell therapy. Finally, advantages of CAR-NK cells over CAR T cells regarding the lack of major toxicity and cytokine release syndrome are emphasized. Furthermore, call attention to better overall survival associated with TIL-NK cells and superior tumor infiltration capacity of TIL-NK cells.
Author Response
We thank the Reviewer for the positive comments regarding our manuscript.
Reviewer 3 Report
Comments and Suggestions for Authors
The topic is interesting and the paper is quite well written. Nevertheless, in my opinion, some parts need to be improved, I have some comments:
1) Recent data supports that the presence of TIL-NK cells is associated with improved overall patient survival and superior tumor infiltration capacity. In summary, this review provides an overview of NK cell expansion methods, the current clinical trial landscape, and the key manipulations aimed at further enhancing NK cell-based adoptive cell therapy. Abstract might be beneficial to include a sentence in the abstract that briefly summarizes the key findings of the study. This can provide readers with a quick overview of the research.
2) Table 2. Major feeder cell-based NK cell expansion methods. Please, improve the legend of this figure.
3) What is the future of NK cell-based therapy? 416 While NK cells hold great promise, challenges remain, including low in vivo persis- 417 tence, negatively impacting their therapeutic efficacy [181]. One major strategy to improve 418 in vivo persistence involves using memory-like NK cells, additionally characterized by su- 419 perior IFN-γ production and cytotoxicity, providing a sustained and potent functional 420 alternative [124]. In addition, using genetically engineered feeder cells, like K562-mbIL21 421 cells, has demonstrated remarkable ex vivo NK cell expansion capacities [84,182]. The conclusion section needs to be improved. It could be interesting to record the aim of the study. It is necessary to clarify the observations of the review and compare them with published literature.
Comments on the Quality of English LanguageThe topic is interesting and the paper is quite well written. This article denotes a great effort by the authors to perform it. Nevertheless, in my opinion, some parts need to be improved, I have some comments, as above reported.
Author Response
1) We have now modified the sentence on TIL-NK presence and better patient outcome, by describing better the results in the studies reporting this finding.
2) As suggested by this Reviewer, we have revised the legend of Table 2 (as well as of Table 1) and hope that the Reviewer will find it improved compared to the original ones.
3) Following this Reviewer suggestion, we have now improved the final section of the manuscript, by summarizing the most important conclusions of the revised literature on different techniques for expanding and improving NK cells for adoptive cell therapy, as well as added a small summary sentence at the end of the text.
Reviewer 4 Report
Comments and Suggestions for Authors
The article provides a good overview of NK cell expansion methods, clinical trial landscape, (although small number of studies) and future of NK-based therapies. The authors approach of beginning from the basics and expanding thereafter can capture wider set of readers. Immune cell redirection is developing into a key immunotherapeutic module which holds promise towards targeting solid tumors as well. This review sheds light (although, not in detail) about various approaches envisaged in this field.
Author Response
We thank the Reviewer for the positive comments reagrding our manuscript.
Round 2
Reviewer 3 Report
Comments and Suggestions for Authors
The authors edited the manuscript and took my suggestions into account. In my opinion this improved the manuscript. I have no further comments.
Comments on the Quality of English LanguageMinor changes of English language are required.